# OpenReview forum: "Keeping LLMs Aligned After Fine-tuning: The Crucial Role of Prompt Templates"
_NeurIPS.cc/2024/Conference — NeurIPS 2024 poster_

### Official Review · Reviewer_JbMw · 2024-06-12

**Soundness:** 3
**Presentation:** 3
**Contribution:** 3
**Rating:** 7
**Confidence:** 5

**Summary:**

This paper proposes a mitigation strategy called "pure tuning, safe testing" to mitigate harmful finetuning issues for LLMs. The strategy is very simple, basically to use a safety system prompt for inference and do finetuning without such a prompt. The core philosophy is that harmful knowledge in the finetuning stage is learned without a safety prompt, but in inference time the the added safety prompt is used and therefore harmful knowledge will not be activated.

**Strengths:**

1. The studied problem -- harmful finetuning for LLMs by itself is important and has raised widespread public interest among the community,  and this paper is one of the early batches of papers to propose a timely analysis and mitigation strategy for the problem.

2. Comprehensive evaluation is conducted to show the effectiveness of the method.

3. The paper is well-written, and the proposed strategy is simple enough to understand, which I think may raise the common interest among the community.

**Weaknesses:**

1. The core issue of PTST is that:  given that the system prompt has changed between finetuning/testing, it does not make sense to me that why the helpfulness is not degraded while the harmfulness is lowered. Both benign helpful knowledge/harmful knowledge are learned with the finetuning system prompt, changing the template in the inference time will simultaneously lower helpfulness/harmfulness in my view.  However, this is not the case in Table 2 (a) and Table 3(a), which indicates that changing the template will not always lower helpfulness (sometimes even increase helpfulness, e.g., CA->CL). I conjecture the reason is that the length of CL prompt is longer, which elicits better helpfulness performance. An explanation for this phenomenon will be appreciated.


2. The observation in Section 4 that mixing safety data can reduce ASR is available in Vlguard Zong et al. [2024]. I understand that this is a concurrent finding, but it would be nice if the authors could mention and discuss this in Section 4.

3. The experimental results are not intuitive enough. Particularly,  I think it is not ideal to use so many prompts (e.g., TV, TA,CV,CA,CL) for comparison. When I am reading  Table 2, I am confused about which one is a safety prompt and which one is not a safety prompt, and therefore, I cannot immediately get the intuition shown by the results.

4. The literature review seems to be comprehensive, but there are a few related works missing. Since (Qi et al, 2024), there are a few mitigation solutions proposed to address the same challenges. I would appreciate it if the authors could appropriately cite and discuss these literature:

------------------Before NeurIPS review cycle------------------

[1] Fine-tuning can cripple your foundation model; preserving features may be the solution https://openreview.net/forum?id=VQ7Q6qdp0P (ICLR2024 template)

[2] Immunization against harmful fine-tuning attacks https://arxiv.org/pdf/2402.16382 （ICLR2024 workshop template）

------------------concurrent------------------

[3] Representation noising effectively prevents harmful fine-tuning on LLMs   https://arxiv.org/pdf/2405.14577 （NeurIPS2024 template）

[4] Lazy Safety Alignment for Large Language Models against Harmful Fine-tuning  httpsImmunization against harmful fine-tuning attacks  https://arxiv.org/abs/2405.18641   （NeurIPS2024 template）

[5] No Two Devils Alike: Unveiling Distinct Mechanisms of Fine-tuning Attacks   https://arxiv.org/pdf/2405.16229 （NeurIPS2024 template）

[6] Safe LoRA: the Silver Lining of Reducing Safety Risks when Fine-tuning Large Language Models  https://arxiv.org/pdf/2405.16833v1 （NeurIPS2024 template）

[7] A safety realignment framework via subspace-oriented model fusion for large language models https://arxiv.org/pdf/2405.09055 （Elsivier Journal template, first available May, 2024）

[8] Navigating the Safety Landscape: Measuring Risks in Finetuning Large Language Models https://arxiv.org/abs/2405.17374 （NeurIPS2024 template）

I am aware that some of the listed work is concurrent work (e.g., con-current submissions to NeurIPS 2024). However, it is encouraged to also cite and discuss them, because that will be beneficial for the development of the research field (but the authors should at least cite those existing works that appeared before the NeurIPS2024 review cycle).

5. Baselines for comparison are lacking.  As there are already a few mitigation strategies for harmful finetuning issues, I suggest the authors add one or two baselines, e.g., Vaccine[Huang et al., 2024] for comparison.

**Questions:**

See the weakness part. I still have questions regarding the results:

In table 2 (b) (c), why changing CL->CV will increase the harmfulness while CL->CA will decrease it?

**Limitations:**

The authors have discussed the limitations, but I suggest the authors discuss the potential impact that the helpfulness will be lower with changing the template (see the first weakness), although this is not that apparent per the authors' experimental results.

Overall, I believe that the idea of changing the template in finetuning/testing should reduce the risk of harmful finetuning, but of course, should come with the potentially negative impact that downgrades the finetune performance. I am willing to increase the score, as long as my detailed questions are answered and the limitation is clearly discussed in the paper.

---

> ### Author Rebuttal · Authors · 2024-08-07
>
> We sincerely thank the reviewer for the positive and constructive feedback. Below, we address the reviewer’s questions.
>
> **Q1** (rephrased): It would be appreciated if the authors could provide an explanation for why using different templates does not simultaneously lower the helpfulness while the harmfulness is lowered. Is it because longer prompts such as CL can elicit better helpfulness performance?
>
> **A1:** Thanks for the reviewer’s suggestion. We respond to this question in our general response. Our hypothesis is that there is some compositional generalization ability arising from the pretraining and alignment phases (prior to the fine-tuning phase we study). Understanding this would be an interesting future direction.
>
> The length of the prompt may also be a factor, but Figure 3(b) shows that a shorter version of CL (llama-short) can achieve an even better helpfulness than CL.
>
> -----------------------
>
> **Q2:** The VLGuard paper (Zong et al., 2024) also observed that mixing safety data can reduce ASR. It would be nice if the authors could mention and discuss it in Section 4.
>
> **A2:** Yes, the VLGuard paper curated a vision-language safety instruction-following data and showed that mixing this data during fine-tuning can mitigate safety issues. In our NeurIPS submission, we have already included a discussion on this work in Section 5 (Related Works). We will also cite this paper in the first paragraph of Section 4.
>
> Meanwhile, we want to clarify that the main point of our Section 4 is not to repeat the point that mixing some safety data mitigates the safety issue, but to show that adding the PTST strategy on top of mixing the safety data can further enhance the safety performance.
>
> -----------------------
>
> **Q3** (rephrased): Table 2 includes too many prompt templates, making it hard for readers to see which ones contain safety prompts.
>
> **A3:** Below, we respond to the review assuming the reviewer was talking about Table 1, since Table 2 only contains 3 templates.
>
> We apologize for any inconvenience caused by the presentation of Table 1. In the next version, we will add a “\*” to the prompt template (CL) that contains a safety prompt. In the current version, blue cells stand for training and testing with the same templates and orange cells stand for PTST. We will revise the caption to explain the meaning of these colors and the “\*” symbol to be added.
>
> -----------------------
>
> **Q4:** The literature review seems to be comprehensive but still some related works are missing.
>
> **A4:** Thanks for listing so many related works before and concurrent to our work. We will definitely cite and discuss them in the next version of our paper!
>
> -----------------------
>
> **Q5:** I suggest the authors add one or two baselines for comparison, e.g., Vaccine (Huang et al., 2024).
>
> **A5:** Thanks for the suggestion. We added Self-Reminder (SR) and In-context Defense (ICD) as two lightweight baselines. Please see the general response above for the results.
> Comparing with Vaccine would also be interesting, but we want to note that this method is orthogonal to our PTST. Vaccine is a robust alignment method that works in the alignment phase (prior to the fine-tuning phase we study). It can make the model’s safety more robust to fine-tuning, but even if Vaccine works well, one can still add PTST during the fine-tuning phase to enhance the model’s safety even further. In our new experiments, we choose to focus on comparing with baselines that work in the fine-tuning phase.
>
> -----------------------
>
> **Q6:** In Table 1(b)(c), why does changing CL->CV increase harmfulness while changing CL->CA decreases it?
>
> **A6:** Thank the reviewer for carefully reviewing our results. First, we would like to clarify our main point in Table 1:
> 1. There exist many cases where using different training and testing templates leads to lower ASR than using the same one, while still improving the model on helpfulness. This can be seen by comparing diagonal entries and off-diagonal entries in Table 1.
> 2. However, not all off-diagonal entries are safer than diagonal entries, just as the reviewer points out.
> 3. Among the off-diagonal entries, the PTST strategy consistently succeeds. That is, when the training template does not have a safety prompt (e.g., TV, TA, CV, CA) and the testing template has a safety prompt (e.g., CL in Table 1 and three other such templates we tried in Figure 3), the ASR is consistently lower than training with the same prompt template and the helpfulness consistently improves. This can be seen by comparing the diagonal entries and the orange-colored off-diagonal entries in Table 1.
>
> Regarding the reviewer’s question on the results of CL->CV and CL->CA, we would like to point out that these strategies are not PTST, and we do not claim that they will consistently reduce or increase the ASR. Besides PTST, there may be other strategies that consistently work, and exploring these other strategies can be an interesting future direction.

---

> ### Comment · Reviewer_JbMw · 2024-08-07
> **Thanks for the rebuttal**
>
> For Q3, I am referring to Table 2. After one month, I again forgot how to interpret Table 2. I think it is generally not a good idea to use the table showing training/testing combination. Dumb people (like me)  may not easily interpret your results (a lot of abberviation for prompt template make situation worse). Perhaps it will better to provide at least one easier to interpret table to show your performance?
>
> It is also interesting to see that the method can be combined with alignment stage solution, e.g., Vaccine. Could you discuss a little bit about this potential combination in the next version of the paper? (perhaps in the last section when you talk about conclusion and future direction)?

---

> > ### Author Response · Authors · 2024-08-10
> >
> > We would like to thank the reviewer's suggestion. In the next version, we will definitely try our best to present Table 2 in a way that is more interpretable to readers. We will also expand our discussion on potential combinations with the alignment stage solution in the last section (Conclusions) and cite related papers (e.g., Vaccine).

---

> > > ### Comment · Reviewer_JbMw · 2024-08-14
> > > **Thank you for the addtional results, I have slightly increase my score!**
> > >
> > > As my initial score is 6, and the authors have provided addtional experiments to justify my concern. I decided to rasie the score to 7.

---

### Official Review · Reviewer_1poN · 2024-06-17

**Soundness:** 3
**Presentation:** 3
**Contribution:** 3
**Rating:** 6
**Confidence:** 4

**Summary:**

This paper shows that the prompt templates used during fine-tuning and inference play a crucial role in safety alignment. Then, the authors propose to fine-tune models without a safety prompt, but include it at test time (user inference), which is counter to intuition. The authors demonstrate their method in the following experiments: when using the same prompts during training on GSM8K and testing, attack success rate increases for a Llama-2-Chat model (the authors considered 5 different prompts). The authors also show the same trend across models GPT-3.5 Turbo, Mistral-7B-Instruct-v0.2, and Llama-2-7b-chat, and across datasets ChatDoctor and OpenOrca.

**Strengths:**

This is a paper that points out a new direction in safety alignment for fine-tuning language models. The paper is written very clearly, with a novel method and supporting experimental results.

**Weaknesses:**

Improvements to this paper can be made from the following aspects: (1) there still seems to be noise in the experiment results, although that does not take away the novelty in proposing the PTST approach, (2) there should be more discussion about implications of PTST

**Questions:**

I have two specific questions and one general question:

Q1. This is a question regarding the experiment results, which I find convincing but nevertheless flawed. In Table 1, (b) shows the trend that this paper is arguing for, specifically that training and testing on the same prompt template makes attacking easier. However, I question whether that is generally the case? The trainTA-testCV entry suffers the same ASR as trainCV-testCV in (b), and the trainCV-testTV entry suffers even higher ASR than trainTV-testTV in (d). I don't think these outliers invalidate the general trend of PTST results, but I still question how universally applicable PTST will be, in the sense that it is unclear whether there is a "PTST prompt" that will perform well under all scenarios? Perhaps there is a hidden confounder at play here?

Q2. Is there a particular reason that TV and TA is not included in the further experiments like GPT-3.5 (judging from Table 1, there are certain cases when TV and TA perform the best)?

Q3. This is a high-level question about the message that this paper is sending. Throughout experiments in this paper, it seems like there is always a tradeoff between helpfulness and ASR. Philosophically, is that really the case? I personally like the paper and believe that PTST is an interesting new direction of research, but I wonder whether a sufficiently intelligence machine still needs to give up either helpfulness or safety? I think the paper is lacking in discussion about the implications of PTST, and how the PTST method might inspire future papers to explore the direction of safety aligned fine tuning.

**Limitations:**

The authors have discussed limitations in the conclusion section.

---

> ### Author Rebuttal · Authors · 2024-08-07
>
> We sincerely thank the reviewer for acknowledging that the paper is clearly written and that the proposed PTST strategy is novel. Below, we address the reviewer’s questions.
>
> **Q1:** This paper argues that training and testing on the same prompt template makes attacking easier. However, the trainTA-testCV entry suffers the same ASR as trainCV-testCV in Table 1(b), and the trainCV-testTV entry suffers an even higher ASR than trainTV-testTV in 1(d).
>
> **A1:** Thank the reviewer for carefully reviewing our results. In fact, we would like to clarify that we do NOT claim that using different training and testing templates is *always* safer than training with the same template, but there exist many such cases, and PTST is a particular case where such a phenomenon consistently happens. More specifically, our claim is the following:
> 1. There exist many cases where using different training and testing templates leads to lower ASR than using the same one, while still improving the model on helpfulness. This can be seen by comparing diagonal entries and off-diagonal entries in Table 1.
> 2. Among these cases, the PTST strategy (described in the gray box on Page 2) consistently succeeds. That is, when the training template does not have a safety prompt (e.g., TV, TA, CV, CA) and the testing template has a safety prompt (e.g., CL in Table 1 and three other such templates we tried in Figure 3), the ASR is consistently lower than training with the same prompt template and the helpfulness consistently improves. This can be seen by comparing the diagonal entries and the orange-colored off-diagonal entries in Table 1.
>
> We will make this point more clear in the next version of our paper.
>
> -------------------
>
> **Q2:** Is there a particular reason that TV and TA is not included in the further experiments like GPT-3.5
>
> **A2:** Yes, the main reason is that we are fine-tuning GPT-3.5 with OpenAI’s APIs, but these APIs do not support TV and TA and only accept chat-mode data represented as a list of system, user and assistant messages.
>
> -------------------
>
> **Q3:** Throughout experiments in this paper, it seems like there is always a tradeoff between helpfulness and ASR. Philosophically, is that really the case? I wonder whether a sufficiently intelligence machine still needs to give up either helpfulness or safety? The paper is also lacking in discussion about the implications of PTST, and how the PTST method might inspire future papers to explore the direction of safety-aligned fine tuning.
>
> **A3:** Thank the reviewer’s suggestion on adding more high-level discussion. We will definitely do in the next version of the paper.
> For existing models, Table 2 and Figure 2(b) showed results on GPT-3.5 Turbo, which is more intelligent than Llama 2-Chat, but still GPT-3.5 Turbo suffers from safety issues after custom fine-tuning with benign data. In this situation, PTST is helpful to mitigate such issues.
> Although we cannot perfectly predict the future, we believe that a model that is good at both helpfulness and safety won’t appear just by adding more data, as discussed in many existing papers [1, 2]. We indeed need better designs of pretraining and alignment methods to achieve this goal.
> For designing better pretraining and alignment methods, it would be helpful if one could know better how the model is going to be used after fine-tuning. Our paper points out the PTST principle for fine-tuning the model after the alignment phase, and an interesting future direction is to identify important factors in pretraining and alignment methods that make PTST work for current models, and improve these methods to make PTST work better in future models. Please also see our general response for a more detailed discussion.
>
> [1]. Jailbroken: How Does LLM Safety Training Fail? https://arxiv.org/abs/2307.02483
>
> [2]. Sleeper Agents: Training Deceptive LLMs that Persist Through Safety Training https://arxiv.org/abs/2401.05566

---

> > ### Comment · Reviewer_1poN · 2024-08-11
> > **I recommend acceptance**
> >
> > Many thanks to the authors for carefully responding to my questions and answering all of them. I have carefully evaluated all the reviews and responses. Seeing that the only reject score is given by Reviewer jeuL and their questions have also been sufficiently answered, I would happily recommend this paper for acceptance in its current state.

---

### Official Review · Reviewer_SwBC · 2024-07-01

**Soundness:** 3
**Presentation:** 3
**Contribution:** 3
**Rating:** 6
**Confidence:** 4

**Summary:**

This paper addresses a critical issue, i,e., LLMs' loss of safety after being fine-tuned. The authors pay their attention to the prompt templates used during fine-tuning and testing, which leads to the main observation that fine-tuning with the valina template and testing with the safe template yields the best robustness.

**Strengths:**

(1) Understanding the effect of fine-tuning on the LLM safety through prompt templates is novel.

(2) The PTST strategy shows promising performance gains when compared with the common strategy where a template is consistently used.

(3) The authors conducted experiments on several templates, models, and datasets.

**Weaknesses:**

(1) The authors leave the understanding of the PTST strategy to future work and very limited discussion on the underlying mechanism of PTST can be found. Although it might be hard to develop a rigorous theory explaining the strategy, I still feel it necessary for the authors to at least propose some hypotheses and try to verify them with concrete experiments.

(2) There are cases when the helpfulness of models is notably decreased if we adopt the PTST rule, such as (TV, CL) and (TA, CL) for Llama-7B.

(3) Some lightweight defenses such as Self-Reminder [1] and ICD [2] can be incorporated into the (CL, CL) training scheme, which will serve as good baselines for PTST. Comparison with safeguarding algorithms can help readers better understand the significance of PTST.

[1] Defending ChatGPT against jailbreak attack via self-reminders; Xie et.al; Nature

[2] https://arxiv.org/abs/2310.06387

**Questions:**

(1) PTST seems to be a general principle to follow when fine-tuning aligned LLMs and the templates considered are restricted to several existing ones. I am wondering whether this principle can help us design better prompt templates for fine-tuning.

**Limitations:**

Please see weakness.

---

> ### Author Rebuttal · Authors · 2024-08-07
>
> We sincerely thank the reviewer for acknowledging the novelty and promising performance of PTST. Below, we address the reviewer’s questions.
>
> **Q1:** I feel it necessary for the authors to at least propose some hypotheses on the underlying mechanism of PTST and try to verify them with concrete experiments. I also wonder whether this principle can help us design better prompt templates for fine-tuning.
>
> **A1:** Thanks for the reviewer’s suggestion. We have responded to this question in our general response.
> * Our hypothesis is that there is some compositional generalization ability arising from the pretraining and alignment phases (prior to the fine-tuning phase we study). Understanding this would be an interesting future direction.
> * The PTST principle may be helpful for designing better prompt templates. It recommends searching for new training templates without mentioning safety and for new testing templates among those emphasizing safety.
>
> ----------------------
>
> **Q2:** There are cases when the helpfulness of models is notably decreased if we adopt the PTST rule, such as (TV, CL) and (TA, CL) for Llama-7B.
>
> **A2:** We would like to thank the reviewer for carefully reviewing our results. It is true that the helpfulness of the models with training and testing templates TV -> CL and TA -> CL are notably lower than those with the same training and testing template. However, we want to note the following two points:
> * Still, the helpfulness is improved after fine-tuning according to the PTST rule. E.g., GSM8K accuracy is 15.31 under No FT -> TV and 6.52 under No FT -> CL, but TV -> CL gives 23.76. This suggests that PTST consistently improves the helpfulness upon the model without fine-tuning, though in some cases PTST may lead to worse helpfulness than the model fine-tuned and tested with the same template.
> * Whether PTST can lead to comparable helpfulness as using the same training and testing templates can depend on the model. For example, on Mistral-7b-Instruct-v0.2, both TV -> CL and TA -> CL give comparable helpfulness improvement as using the same training and test templates (see Table 6). Thus, an interesting future direction is to explore whether we can improve the aligned model in the alignment or even the pretraining phase to make the helpfulness improvement more robust to the change of template. See also the general response above for a detailed discussion.
>
> ----------------------
>
> **Q3:** Some lightweight defenses such as Self-Reminder [1] and ICD [2] can be incorporated into the (CL, CL) training scheme, which will serve as good baselines for PTST. Comparison with safeguarding algorithms can help readers better understand the significance of PTST.
>
> **A3:** Thanks for the suggestion. We added Self-Reminder (SR) and In-context Defense (ICD) as two other lightweight baselines. Please see the general response above for the results.

---

> > ### Comment · Reviewer_SwBC · 2024-08-08
> > **Thanks for the rebuttal**
> >
> > Hi,
> >
> > I appreciate the authors' effort in conducting the extra experiments and find the results helpful.
> >
> > However, the authors still fail to give any experiment/theory-grounded explanation for the effectiveness of PTST even though three out of four reviewers have raised the concern. From my perspective, it is simply not enough to convince the community with repeated experiments on model A/B/C + setting D/E/F. Without such rigorous verification (or at least some basic attempts), I am always doubtful about the correctness of the proposed rule on new models (e.g. Llama-3-Instruct), new datasets (e.g. Function Calling Extended), new fine-tuning strategies (e.g. QLoRA), and new chat templates (e.g. those used by the QWen model herds or templates in languages other than English.)
> >
> > In summary, I will keep my score since Q1 is not well addressed, and I personally do not favor recommending the paper as an oral/award paper.

---

### Official Review · Reviewer_jeuL · 2024-07-11

**Soundness:** 2
**Presentation:** 2
**Contribution:** 2
**Rating:** 6
**Confidence:** 3

**Summary:**

This paper discusses the issue of maintaining model consistency after fine-tuning large language models (LLMs). The research team, through extensive experiments, found that the prompt templates used during fine-tuning and inference play a crucial role in maintaining model safety. The paper proposes the "Pure Tuning, Safe Testing" (PTST) principle, which involves not using safety prompts during fine-tuning but incorporating them during testing to significantly reduce the occurrence of unsafe behaviors.

**Strengths:**

1. Through extensive experiments, it is demonstrated that prompt templates are crucial for maintaining safety during both training and testing.
2. The PTST approach is proposed, which improves safety performance.

**Weaknesses:**

1.Why fine-tune on math datasets (gsm8k, Orca-Math) to verify the model's safety? How does the performance compare when fine-tuned on safety-specific datasets, such as Anthropic/hh-rlhf?\
2.The experiments on PTST are insufficient, as they do not adequately compare the effectiveness of the approach with current alignment algorithms such as PPO, DPO, KTO, among others.\
3.This paper proposes the PTST algorithm, but it is a training technique and lacks a certain level of innovation.

**Questions:**

1.It would be interesting to see whether the approach also scales to more datasets, such as hh-rlhf, or a combination of GSM8k and hh-rlhf for mixed training.\
2.Could you explain what the core contributions of the PTST algorithm are? How does it differ from algorithms like DPO?\
3.How does the performance of PTST compare to aligner[1] on larger-scale datasets?\
[1]Ji J, Chen B, Lou H, et al. Aligner: Achieving efficient alignment through weak-to-strong correction[J]. arXiv preprint arXiv:2402.02416, 2024.

**Limitations:**

1.Is the PTST algorithm still effective with an increasing amount of data or the introduction of mixed datasets?\
2.Lacks comparison with other methods for improving LLM safety.(Aligner,DPO...)

---

> ### Author Rebuttal · Authors · 2024-08-07
>
> We sincerely thank the reviewer for reviewing our paper and for acknowledging our experiments as extensive. However, we’d like to point out that the reviewer’s main comment (1, 2 below) under “weakness” (as well as Question 2, Question 3 and Limitation 2) suggests they **may not** have absorbed the core idea of our paper.
>
> > 1. Why fine-tune on math datasets (gsm8k, Orca-Math) to verify the model's safety? How does the performance compare when fine-tuned on safety-specific datasets, such as Anthropic/hh-rlhf?
> > 2. The experiments on PTST are insufficient, as they do not adequately compare the effectiveness of the approach with current alignment algorithms such as PPO, DPO, KTO, among others.
>
> The starting point of our paper is that often end-users try to improve the capabilities of aligned models  (such as llama chat) on specialized tasks like GSM8K. This fine-tuning leads to a loss of safety alignment  (i.e., a rise in unsafe responses) if one follows the common practice that uses the same training and testing prompt templates, as discussed in the introduction (Section 1; see also Appendix C). Our paper proposes a lightweight trick (called PTST) that avoids such a drastic loss of alignment during post-alignment fine-tuning (see, e.g., Table 1 for safety with/without our method).
>
>
> ### Clarification of Misunderstanding
> The reviewer asks for the comparison of PTST and alignment algorithms in Weakness 2, Question 2, Question 3 and Limitation 2 and for experiments on the human preference dataset hh-rlhf in Weakness 1 and Question 1. However, PTST and alignment algorithms are used in different stages of LLM training and deployment. As highlighted in Sections 1 and 2, our paper studies the scenario where a model owner fine-tunes an **already aligned** LLM to enhance a certain helpfulness metric, e.g., fine-tune on GSM8k to improve math capabilities, measured by the accuracy on the GSM8k test set. PTST helps the fine-tuned LLM retain the safety attribute previously acquired during the alignment training. By contrast, alignment algorithms, e.g., PPO, DPO, KTO, are used to align base LLMs, e.g., Llama 2,  with human preferences prior to the fine-tuning process we investigate.
>
> We believe this clarification already resolves the reviewer’s questions raised in Weaknesses 1, 2, Questions 1, 2, 3, and Limitation 2. Below we respond to the other questions.
>
> ### Other Concerns
> **W3:** This paper proposes the PTST algorithm, but it is a training technique and lacks a certain level of innovation.
>
> **A:** We argue that PTST is a non-trivial training technique. A common practice for fine-tuning is to use the same prompt templates for training and inference to maximize the downstream performance. By contrast, PTST encourages safety by creating a distribution shift in training and inference while still maintaining the downstream performance. Furthermore, PTST is lightweight and easy to implement, allowing for further refinement of aligned LLMs with significantly reduced safety concerns.
>
> **L1:** Is the PTST algorithm still effective with an increasing amount of data or the introduction of mixed datasets?
>
> **A:** Yes. As detailed in Section 3.3 and Appendix F, besides GSM8k, which contains 8k samples, we also conducted experiments with larger datasets, including ChatDoctor with 100k samples and a 600k-sample subset of OpenOrca. I. In Section 4, we fine-tuned the models on a mixture of GSM8k and safety data. PTST consistently maintains safety across all these settings.

---

> > ### Comment · Reviewer_jeuL · 2024-08-11
> >
> > Thanks to the authors for the detail answer. In general, The authors have addressed my main concerns, I increase my rating to 6 (from 4 to 6).

---

### Author Rebuttal · Authors · 2024-08-07

We sincerely thank all the reviewers for their time and effort in reviewing our paper. Below we address some common questions that are raised by more than one reviewer.

**Q1:** Can the authors provide some discussion on why PTST works and how this method might inspire future explorations? (by SwBC, 1poN, JbMw)

**A1:** Our hypothesis is that there is a certain form of compositional generalization when the language model learns to handle user instructions and safety prompts. More specifically, during the pretraining and alignment phases, the model has to learn to perform the following two skills (or tasks):
* S1: Given a user instruction, write a helpful response;
* S2: Given a safety prompt and a user instruction, determine whether it is harmful. If yes, write an appropriate response to refuse to answer it.

Further, the model has to learn how to compose these two skills together:

* S1+S2: Given a safety prompt and a user instruction, do S2 first; if it turns out that the instruction is not harmful, do S1.

We speculate that these two skills become quite “disentangled” in LLMs after they see a massive amount of data during the pretraining and alignment phases. That is, LLMs do S1 or S2 in task S1+S2 in a similar way as if they are doing S1 or S2 alone. Then it makes sense that fine-tuning S1 can lead to helpfulness improvement in both S1 and S1+S2. This corresponds to the case of PTST, since we do not add safety prompts during fine-tuning but add one during testing.

Understanding how this compositional generalization and disentangled skills emerge in LLMs can be an interesting future direction. A possible way to deepen such an understanding is to pretrain and align LLMs in a fully-controlled setting, and do ablation studies to see which part of training contributes the most to the disentanglement between S1 and S2. An example of this research style is [this paper on the physics of LLMs](https://arxiv.org/abs/2309.14316), which explains interesting phenomena about knowledge storage in LLMs in a fully-controlled setting with synthetic data. We believe this is a possible way to go, but due to the time limit and resource constraints, we cannot provide any experiment results yet.

For future research, it would be interesting to explore better pretraining and alignment strategies so that PTST can work even better in the custom fine-tuning phase after alignment. This would require a lot of speculation and understanding of which part of training contributes the most to the compositional generalization mentioned above.

-----------

**Q2:** Can the authors provide more comparisons with some baseline methods? (by SwBC, JbMw)

**A2:** Following the reviewer SwBC’s suggestion, we added Self-Reminder (SR) and In-context Defense (ICD) as two other lightweight baselines. The following table shows the ASRs on DirectHarm4 when using training and testing templates. Each row represents a training template, and each column represents a testing template. Due to limited time, the newly added experiments are only conducted with a single seed. The other entries in the table below are directly copied from Table 1.

| train \ test  | CV | CA | CL | SR | ICD |
|-------------:|------|----|-----|-----|-----|
| **CV**             | 11.00 | 20.50 | 1.08 | 0.00 | 0.25 |
| **CA**             | 8.08 | 46.42 | 1.00 | 0.75 | 2.00 |
| **CL**             | 6.83 | 18.92 | 18.08 | 7.00 | 1.00 |
| **SR**             | 11.50 | 39.75 | 8.00 | 20.25 | 3.00 |
| **ICD**            | 21.00 | 33.75 | 4.25 | 3.25 | 26.75 |

These results show that baseline methods using SR or ICD for both training and testing lead to high ASRs. Consistent with our paper’s main claim, training with prompt templates that do not emphasize safety (CV, CA) and testing with templates that contain safety prompts (CL, SR,  ICD) generally lead to very low ASR.

---

### Decision · Program_Chairs · 2024-09-25

**Decision:**

Accept (poster)

**Comment:**

The paper investigates the problem of maintaining the safety alignment of large language models (LLMs) after fine-tuning. It introduces the "Pure Tuning, Safe Testing" (PTST) strategy, which suggests fine-tuning LLMs without safety prompts but incorporating them during testing. The paper presents evidence that this strategy significantly reduces the likelihood of unsafe behaviors while maintaining performance on downstream tasks. The approach is tested across multiple LLMs, including Llama 2-Chat, Mistral 7B Instruct, and GPT-3.5 Turbo, on datasets such as GSM8K, ChatDoctor, and OpenOrca.

The reviewers unanimously agreed on the importance of the problem being addressed, and the proposed PTST strategy is a novel and counterintuitive approach that introduces an intentional distribution shift to maintain alignment. The paper is well-supported by comprehensive experiments across several LLMs and datasets, providing strong evidence for the effectiveness of the proposed strategy.

Some reviewers pointed out that while the PTST strategy is interesting, the paper could have provided deeper theoretical justification for why this approach works. And generalizability of the PTST strategy across different types of fine-tuning tasks and LLMs is also raised. The authors provided detailed responses to the reviewers’ concerns, adding additional explanations and justifications. They provided explanations on the theoretical motivations behind the approach and provided further experimental data to support generalizability issue. The reviewers generally appreciated these efforts, although some concerns about the depth of the theoretical analysis remained. While there are areas where the paper could be strengthened, particularly in theoretical justification and broader generalization, based on the overall contribution and experimental validation, I propose acceptance.